# Seeing the part and knowing the whole: Object-Centric Learning with Inter-Feature Prediction

## Abstract

Humans can naturally decompose scenes into understandable objects, resulting in strong visual comprehension ability. In light of this, Object-Centric Learning (OCL) seeks to explore how to construct object-level representations by encoding the information of objects in the scenes into several object vectors referred to as 'slots'. Current OCL models rely on an auto-encoding paradigm that encodes the image feature into slots and reconstructs the images by composing the slots. However, merely reconstruction objectives do not guarantee that each slot exactly corresponds to a holistic object. Existing methods often fail when objects have complex appearances because the reconstruction objective cannot indicate which pixels should be assigned to the same slot. Therefore, additional regularization based on a more general prior is required. For this purpose, we draw on the gestalt ability that humans tend to complete a broken figure and perceive it as a whole, and propose Predictive Prior that features belonging to the same object tend to be able to predict each other. We implement this prior as an external loss function, demanding the model to assign features that can predict each other to the same slot, and vice versa. With experiments on multiple datasets, we demonstrate that our model outperforms previous models by a large margin in complex environments where objects have irregular outlines and intense color changes, according to various tasks including object discovery, compositional generation, and visual question & answering. Visualization results verify that our model succeeds in discovering objects holistically rather than dividing them into multiple parts, proving that Predictive Prior gives a more general object definition. Code is available at https://anonymous.4open.science/r/PredictivePrior-32EF.

## 1 Introduction

The world is highly compositional. Individual objects make up visual scenes. Humans have developed object vision that allows for understanding complex visual scenes by breaking them down into individual objects (Kahneman et al., 1992). Such structural representations with objects as the base unit are crucial for many important visual properties, such as systematic generalization (Kuo et al., 2021), compositional generation (Singh et al., 2021), and visual reasoning (D'Amario et al., 2022). However, extracting objects from unstructured RGB pixels is unnatural for neural networks. To address this problem, Object-Centric Learning (OCL) is proposed to include the concept of objects in the network design explicitly. Formally, object-centric models are trained to represent images with a set of latent object vectors which are often referred to as 'slots' (Greff et al., 2019; Locatello et al., 2020; Burgess et al., 2019), where each slot contains the information of an individual object.

Current mainstream OCL models follow a slot-based auto-encoding structure (Locatello et al., 2020) that encodes image features into several slots and reconstructs images with these slots. However, merely auto-encoding objective does not guarantee the correspondence between slots and objects. Since objects do not naturally emerge from pixels, the models need prior information to discover object regions. Previous work (Singh et al., 2021; Seitzer et al., 2022) observed that models that reconstruct raw RGB pixels rely on color bias and tend to assign regions with gentle color change to a slot. Although color trends inside objects are commonly gentle, this doesn't always work. Instead, the simple color bias may lead to inferior binding in more complex scenarios. For example, in Fig.3,

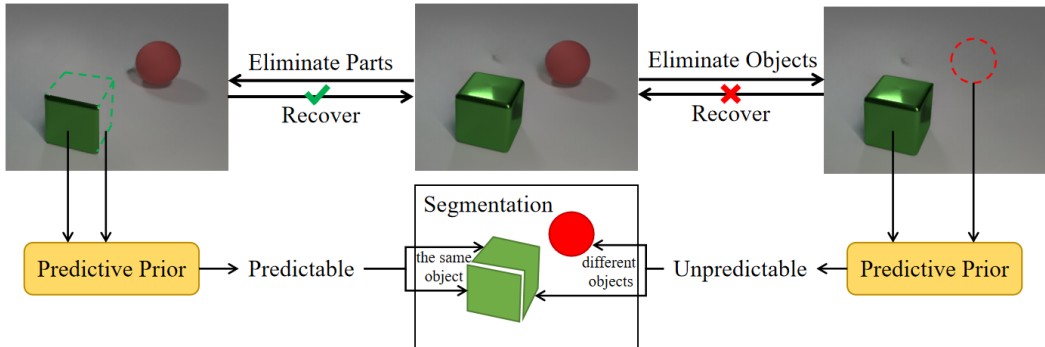

Figure 1: **From gestalt ability to Predictive Prior.** Human vision can complete objects: If we eliminate object parts (e.g., faces of the cube) from the image, it is feasible to recover the eliminated part with the remaining part. However, when we eliminate a holistic object (e.g., the red ball), its information is difficult to infer from its surroundings. This helps us distinguish between objects and parts, i.e. image features belonging to the same object tend to be able to predict each other. Based on this property, we propose Predictive Prior to predict among image features, requiring the model to assign features with stronger prediction relationships to the same slot.

we observe that objects are often divided into multiple parts rather than being identified holistically in MOVi-C. In addition, in the models that introduce DINO features (Seitzer et al., 2022) or diffusion decoders (Jiang et al., 2023), slots may be bound to fixed spatial locations rather than objects. The examples in Fig.3 show the case in Super-CLEVR (Li et al., 2023) and PTR (Hong et al., 2021) where these models can only provide trivial segmentations. Therefore, more general priors are required to capture object representations across various scenarios.

In this paper, we draw on human's gestalt ability to propose Predictive Prior that distinguishes objects according to the prediction relationship between image features. It has been pointed out that humans tend to complete broken figures in the process of visual perception (Spelke, 1990; Wagemans et al., 2012). Similarly, the success of neural networks in processing occluded objects (Xie et al., 2022; Ozguroglu et al., 2024) shows that self-supervised features have the potential to infer about invisible object parts. Based on this fact, we propose Predictive Prior, that is, image features belonging to the same object tend to be able to predict each other and vice versa. For example, as is shown in Fig.1, we can use one face of a cube to infer the position and appearance of other faces, but we cannot infer whether there is a sphere nearby. Predictive Prior promotes the models to assign features with stronger prediction relationships to the same slot, thus discovering individual objects. Specifically, we train a prediction network that uses a given feature to predict features in other spatial locations. The prediction network gives a quantitative constraint that if two features can predict each other well, they should be assigned to the same slot. Therefore, when training OCL models, we construct supervision on the object masks by selecting features and mask pairs with randomly sampled spatial locations. The mask pairs are supervised according to the Predictive Priors between the features: high Predictive Priors indicate similar masks, and vice versa.

We evaluate our proposed method on multiple datasets including MOVi-C (Greff et al., 2022), Super-CLEVR (Li et al., 2023), and PTR (Hong et al., 2021) that cover various multi-object scenarios composed of complex objects such as vehicle models, furniture, or realistic objects. The shapes and colors of the objects in these datasets are irregular, making it challenging to identify objects holistically. We evaluate the models on various tasks to demonstrate the model's improvement in object-centric representations. First, we focus on the unsupervised object discovery task where we show that Predictive Prior brings about clearer object definition, solving the problem that previous methods tend to divide objects into multiple parts in complex scenes. Second, we introduce the compositional generation task to verify that the model can store objects in the slots holistically and compose them into new scenes. Our model generates clear images without obtrusive object parts. Third, we adopt the visual question & answering task to demonstrate that the slots contain high-level semantics of their corresponding objects. We verify that the VQA performance has a strong correlation with the object-centric representations, and the improvement is significant in the problem of the attribute of a target object. Finally, we analyze multiple priors proposed by previous segmentation research in the ablative experiment and verify the superiority of Predictive Prior.

To sum up, current object-centric models are poor at processing complex objects due to the lack of general prior. The model's structural inductive bias, e.g., color bias, is not sufficient to identify objects with irregular appearances. Therefore, additional supervision is required to learn object-centric representations. For this purpose, we propose the Predictive Prior that utilizes the prediction relationship between self-supervised features to judge whether these features come from the same object and construct a loss function based on Predictive Prior. We conduct experiments over multiple datasets and tasks and show that Predictive Prior largely enhances the object-centric representations.

## 2 RELATED WORK

**Object-Centric Representation Learning.** OCL attempts to perceive environments in terms of object-based elements. Mainstream OCL methods follow an auto-encoding paradigm that first encodes input signals into several slots and reconstructs input with these slots. Earlier works, including IODINE (Greff et al., 2019), MONet (Burgess et al., 2019) and GENESIS (Engelcke et al., 2019), accomplish this task by using multiple encoder-decoder structures. Slot-Attention (Locatello et al., 2020) proposed an iterative attention mechanism that allows slots to compete for image segments. Follow-up studies improve the slot-attention-based model from several aspects to adapt OCL to complex scenes. BO-QSA (Jia et al., 2023), I-SA (Chang et al., 2022) and InvariantSA (Biza et al., 2023) focus on query optimization, which uses learnable parameters to initialize slots instead of random sampling. SLATE (Singh et al., 2021) and LSD (Jiang et al., 2023; Wu et al., 2023) attempt to improve the decoder structure, introducing transformer-based and diffusion-based decoders to enhance the model's reconstruction ability. DINOSAUR (Seitzer et al., 2022) proposes to replace the RGB pixel reconstruction objective with the output feature of DINO (Caron et al., 2021).

**Exploring prior knowledge to discover objects.** Slot-based models contain the structural prior that pixels with similar features such as locations and colors tend to be assigned to the same slot (Singh et al., 2021; Seitzer et al., 2022), thus achieving success on simple synthetic datasets (Johnson et al., 2016; Kabra et al., 2019; Karazija et al., 2021) while degrading largely on more complex scenes. In addition, the auto-encoding objective does not indicate object representations, i.e. minimizing reconstruction losses does not necessarily result in better object-centric representations. A promising approach to address this problem is introducing stronger prior knowledge to define and represent objects. LearnToCompose (Jung et al., 2024) enhances the compositional generation ability of object-centric models with the generative prior of diffusion model, making the image generated by slot composition more reasonable. VideoSAUR (Zadaianchuk et al., 2024) leverages inter-frame feature similarity to capture moving objects. Unsupervised segmentation studies (Wang et al., 2023; Hamilton et al., 2022; Lan et al., 2024) also explore several methods to segment objects or semantic classes based on feature similarity between self-supervised pre-training models. In this paper, we will revisit how can self-supervised pre-training models guide object-centric representations.

## 3 METHOD

### 3.1 PRELIMINARY: SLOT-BASED OBJECT-CENTRIC MODEL

Our model follows previous object-centric models and adopts a slot-based structure that contains three components. First, a backbone network $\mathcal{E}_{\text{backbone}}$ encodes the input image $\mathbf{I} \in \mathbb{R}^{3 \times H \times W}$ into image features $\mathbf{F}_{\text{I}} \in \mathbb{R}^{C_I \times H_I \times W_I}$. For CNN-based image encoders, a position embedding is added to $\mathbf{F}_{\text{I}}$. Second, a slot encoder $\mathcal{E}_{\text{S}}$ extracting $K$ slots $\mathbf{S} \in \mathbb{R}^{K \times C_S}$ from $\mathbf{F}_{\text{I}}$. Typically, the slot encoder repeats several times to calculate the attention between $\mathbf{F}_{\text{I}}$ and $\mathbf{S}$ and updates $\mathbf{S}$ with a GRU cell:

$$
\begin{aligned}
\mathbf{A}(\mathbf{F}_{\text{I}}, \mathbf{S}) &= \texttt{softmax}(\frac{\mathcal{K}(\mathbf{F}_{\text{I}}) \cdot \mathcal{Q}(\mathbf{S})^T}{\sqrt{C_S}}, \text{axis} = \mathbf{S}), \\
\mathbf{U} &= \mathbf{A}(\mathbf{F}_{\text{I}}, \mathbf{S})^T \cdot \mathcal{V}(\mathbf{F}_{\text{I}}), \\
\mathbf{S} &\leftarrow \text{GRU}(\mathbf{S}, \mathbf{U}),
\end{aligned}
\tag{1}
$$

where $\mathcal{Q}, \mathcal{K}, \mathcal{V}$ represents linear projections to acquire queries, keys, and values. Finally, a slot decoder $\mathcal{D}_{\text{S}}$ decodes slots into reconstructions $\mathbf{R} \in \mathbb{R}^{3 \times H \times W}$. $\mathcal{D}_{\text{S}}$ also generates an assignment mask $\boldsymbol{\alpha}$ to show which slot each pixel is assigned to. For mixture-based decoders (Watters et al., 2019), we use the object mask generated by the decoder as $\boldsymbol{\alpha}$. For transformer-based decoders

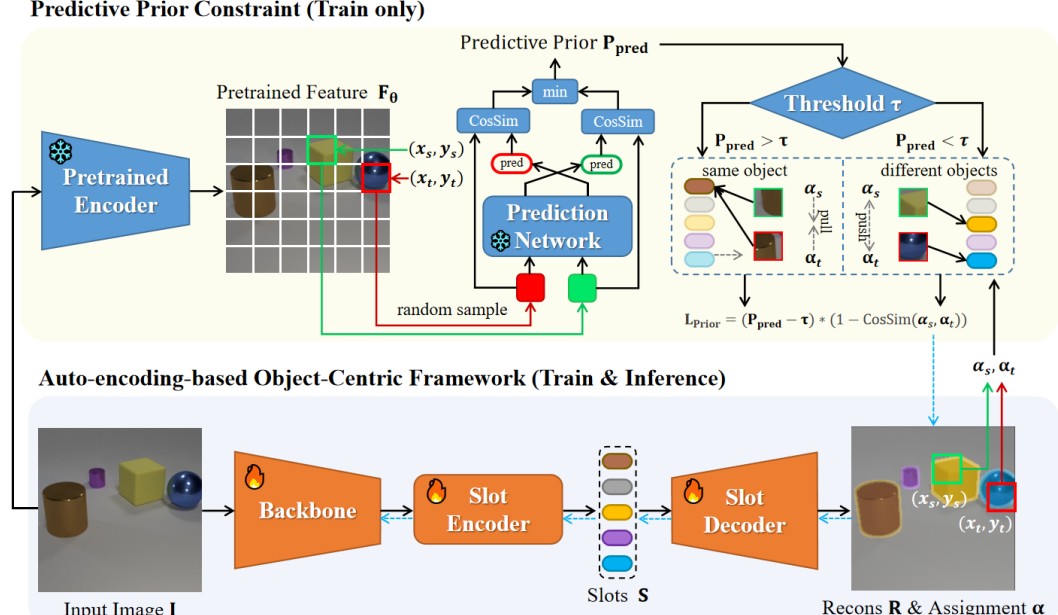

Figure 2: **Object-Centric Learning Framework with Predictive Prior**. The input images pass through two pathways. The first is the auto-encoding path that encodes the images into slots and uses a slot decoder to produce reconstruction and mask for each slot. The second path generates the Predictive Prior that supervises the object masks. A self-supervised encoder first extracts pre-trained features from the images. Feature pairs are sampled with random spatial locations, and the Predictive Prior is computed by letting them predict each other. Masks at the same spatial locations are supervised by the Predictive Prior: they are pulled in when the value of Predictive Prior is higher than a preset threshold, otherwise, they are pulled away.

(Singh et al., 2021), we use the attention map of the last attention layer. Models are optimized by minimizing the reconstruction loss between $\mathbf{R}$ and $\mathbf{I}$. We use L1 loss and perceptual loss (LPIPS) (Zhang et al., 2018) as the reconstruction loss. The training loss is written as:

$$\mathbf{L}_{\text{rec}} := \|\mathbf{R} - \mathbf{I}\|_1 + \text{LPIPS}(\mathbf{R}, \mathbf{I}) \tag{2}$$

## 3.2 PREDICTIVE PRIOR DEFINES OBJECTS

As is discussed in Sec.1, neural networks have the potential to predict object parts with features from other parts. Therefore, we propose Predictive Prior to represent semantic correlations between features. Intuitively, two features belonging to the same object tend to share high mutual information to predict each other. Otherwise, they may contain almost no information about each other.

To achieve this intuition, we design a constraint that promotes the model to assign features that can predict each other to the same slot. Specifically, we first utilize a self-supervised pre-trained model $\mathcal{E}_\theta$, such as DINO, to extract features $\mathbf{F}_\theta \in \mathbb{R}^{C_\theta \times H_\theta \times W_\theta}$ from the images $\mathbf{I}$:

$$\mathbf{F}_\theta = \mathcal{E}_\theta(\mathbf{I}). \tag{3}$$

A prediction network $\mathcal{P}$ is trained to predict between feature pairs: We sample a feature pair $f_s$ and $f_t$ from $\mathbf{F}$ with a pair of random spatial locations $(x_s, y_s)$ and $(x_t, y_t)$. During training, $\mathcal{P}$ takes $f_s$ and $(x_t, y_t)$ as input and output a predicted feature $f_p$. $\mathcal{P}$ is optimized by minimizing the cosine distance between $f_t$ and $f_p$. Formally,

$$
\begin{aligned}
x_s, y_s, x_t, y_t &\sim \texttt{U}(-1, 1), \\
f_s, f_t &= \texttt{Sample}(\mathbf{F}_\theta, (x_s, y_s)), \texttt{Sample}(\mathbf{F}_\theta, (x_t, y_t)), \\
f_p &= \mathcal{P}(f_s, x_t, y_t), \\
\mathbf{L}_{\text{pred}} &= 1 - \texttt{CosSim}(f_p, f_t),
\end{aligned} \tag{4}
$$

where 'U' stands for a uniform distribution, 'Sample$(\mathbf{F}, (x, y))$' stands for the grid sampling operation that sample from $\mathbf{F}$ according to the coordinate $(x, y)$, and 'CosSim' stands for cosine similarity. After trained done, $\mathcal{P}$ is frozen and used to define whether two features belong to the same objects. Unlike the training process, we perform a bi-directional prediction in this stage: for $f_s$ and $f_t$ sampled from $\mathbf{F}$, the prediction network predicts $f_s$ with $f_t$ and $f_t$ with $f_s$, acquiring two prediction similarity $\mathbf{Sim}_{s \to t}$ and $\mathbf{Sim}_{t \to s}$, where the smaller one represents the Predictive Prior $\mathbf{P}_{\mathrm{pred}}$ between $f_s$ and $f_t$:

$$
\begin{aligned}
\mathbf{Sim}_{s \to t} &= \texttt{CosSim}(\mathcal{P}(f_s, x_t, y_t), f_t), \\
\mathbf{Sim}_{t \to s} &= \texttt{CosSim}(\mathcal{P}(f_t, x_s, y_s), f_s), \\
\mathbf{P}_{\mathrm{pred}}(f_s, f_t) &= \texttt{min}(\mathbf{Sim}_{s \to t}, \mathbf{Sim}_{t \to s}).
\end{aligned}
\tag{5}
$$

We adopt the bi-directional prediction setting because $\mathbf{P}_{\mathrm{pred}}$ should be symmetric, i.e., $\mathbf{P}_{\mathrm{pred}}(f_s, f_t) = \mathbf{P}_{\mathrm{pred}}(f_t, f_s)$. In addition, we observe that features from different semantic parts have different difficulties in predicting. For example, background features are generally easier to predict than foreground objects. Therefore, we select the smaller prediction similarity so that a large $\mathbf{P}_{\mathrm{pred}}$ only occurs when both $f_s$ and $f_t$ can predict each other well.

### 3.3 Object-Centric Learning with Predictive Prior

Since the reconstruction loss does not indicate object information, previous OCL models with only auto-encoding optimization objectives highly depend on the matching between model structure and dataset. As a result, they often fail to capture valid object representations in complex scenes.

A crucial observation is that there is a strong correlation between the accuracy of the assignment $\boldsymbol{\alpha}$ and the models' object-centric representations: More accurate $\boldsymbol{\alpha}$ commonly indicates better object representations. Therefore, we use the proposed Predictive Prior to construct a constraint that promotes $\boldsymbol{\alpha}$ to fit holistic objects: We introduce a threshold $\tau$ and require features with a Predictive Prior higher than $\tau$ to be assigned to the same slot and vice versa. Technically, we find that directly attaching the constraint to $\boldsymbol{\alpha}$ may make $\boldsymbol{\alpha}$ hard to optimize. Therefore, to achieve better supervision, we introduce an independent segmentation branch to learn a segmentation mask $\mathbf{M}$, and then use $\mathbf{M}$ to supervise $\boldsymbol{\alpha}$. Specifically, we use a shallow convolutional network to restore the image features $\mathbf{F}_{\mathrm{I}}$ to the resolution of the original image, and then obtain $\mathbf{M}$ through the inner product between the upsampled feature and the slots $\mathbf{S}$. For each image, we randomly sample $N$ pairs of spatial positions $(x_s, y_s), (x_t, y_t)$ and use grid sampling to sample $f_s, f_t$ from $\mathbf{F}_\theta$ as well as $m_s, m_t$ from $\mathbf{M}$. Finally, the model is supervised with

$$
\mathbf{L}_{\mathrm{prior}} := ((\mathbf{P}_{\mathrm{pred}}(f_s, f_t) - \tau) * 10).\texttt{clamp}(-1, 1) * (1 - \texttt{CosSim}(m_s, m_t)) + \|\mathcal{SG}(\mathbf{M}) - \boldsymbol{\alpha}\|_1, \tag{6}
$$

where $\mathcal{SG}(\cdot)$ represents the stop-gradient operation, and $\texttt{clamp}(a, b)$ is a PyTorch-style function that truncates values exceed the $[a, b]$ range to $a$ and $b$. When $\mathbf{P}_{\mathrm{pred}}(f_s, f_t)$ is larger than $\tau$, $m_s$ and $m_t$ should have a cosine similarity that is close to 1, indicating that they are assigned to the same slot. Otherwise, their cosine similarity should be close to 0, representing different assignments. The overall objective is then formulated as

$$
\mathbf{L} = \mathbf{L}_{\mathrm{rec}} + \lambda_{\mathrm{prior}} \mathbf{L}_{\mathrm{prior}}, \tag{7}
$$

## 4 Experiments

**Datasets.** We compare our model with other SOTA object-centric models on MOVi-C (Greff et al., 2022), Super-CLEVR (Li et al., 2023) and PTR (Hong et al., 2021). Each image from MOVi-C contains a random HDRI from Poly Haven as the background and several realistic everyday objects from the Google Scanned Objects (GSO) dataset (Downs et al., 2022). Super-CLEVR and PTR respectively introduce vehicle and furniture models into the CLEVR (Johnson et al., 2016) scene to create more challenging situations. For MOVi-C, we resize the image to $224 \times 224$ resolution. For Super-CLEVR and PTR, we crop the center part of the image and resize the cropped image to $128 \times 128$. The RGB values of all the images are normalized to [-1,1].

**Predictive Prior Computation.** For MOVi-C, we use a DINO pre-trained ViT-Small with patchsize 8 as the encoder $\mathcal{E}_\theta$ to extract pre-trained features $\mathbf{F}_\theta$. For Super-CLEVR and PTR, given that they have a large domain gap with the data used for DINO training, we trained an MAE from scratch with

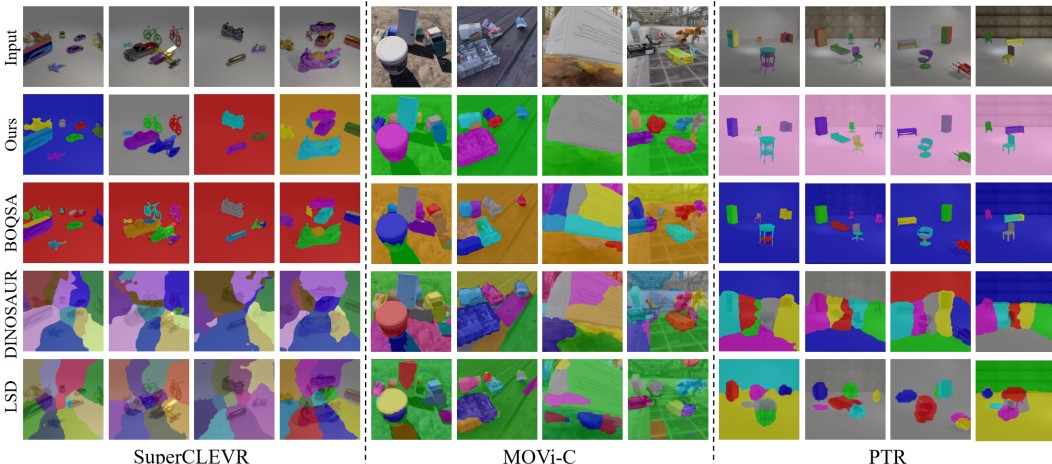

Figure 3: **Unsupervised object discovery results**. Our model shows far greater segmentation accuracy than other methods. It adapts well to different datasets, accurately demarcating the background and identifying the objects holistically.

Table 1: **Unsupervised object discovery comparison.** 'ARI' in the table refers to 'ARI-FG' that only considers foreground pixels. 'mIoU' and 'mBO' are metrics that consider backgrounds. Higher is better for all the metrics.

| Model | MOVi-C | | | Super-CLEVR | | | PTR | | |
|---|---|---|---|---|---|---|---|---|---|
| | ARI | mIoU | mBO | ARI | mIoU | mBO | ARI | mIoU | mBO |
| BO-QSA (Jia et al., 2023) | 58.62 | 44.90 | 46.77 | 70.33 | 57.17 | 57.44 | 66.01 | 63.55 | 65.26 |
| InvariantSA (Biza et al., 2023) | 33.72 | 26.06 | 26.94 | 67.28 | 58.50 | 58.86 | 69.36 | 33.98 | 38.28 |
| DINOSAUR (Seitzer et al., 2022) | 67.82 | 31.16 | 38.18 | 59.52 | 15.29 | 15.59 | 63.80 | 16.16 | 17.57 |
| LSD (Jiang et al., 2023) | 51.98 | 44.19 | 45.57 | 53.05 | 13.15 | 13.38 | 62.22 | 41.16 | 41.34 |
| ours | **74.80** | **59.32** | **60.47** | **86.91** | **60.74** | **61.02** | **70.43** | **70.81** | **71.48** |

images from these datasets to provide $\mathbf{F}_\theta$. The prediction network is a simple 6-layer MLP with a hidden dim of 768. It takes the concatenation of the source features and the position embedding of target features as input and outputs the prediction of target features.

**Object-Centric Framework Setting.** For the object-centric model, we adopt ResNet-34 on Super-CLEVR and PTR, and ViT-Small with patchsize 8 on MOVi-C as the backbone network $\mathcal{E}_{\text{backbone}}$. We choose BO-QSA module (Jia et al., 2023) as the slot encoder $\mathcal{E}_S$. The number of slots $K$ is set to 11 for Super-CLEVR and MOVi-C and 7 for PTR, i.e., the maximum number of objects in an image plus one (for backgrounds). For Super-CLEVR and PTR, we use a mixture-based slot decoder that independently decodes each slot into an RGB reconstruction and an alpha mask, and composes the final reconstruction through alpha blending. For MOVi-C, we use a transformer-based decoder that integrates the information from slots through multiple cross-attention layers. The attention map from the last layer is taken as the object mask.

**Compared models.** As mentioned in Sec.2, previous methods improve the slot-based model from three aspects: improving the slot-attention module, introducing more capable decoder modules, and changing the reconstruction target from RGB pixels to other signals. We choose SOTA models from these three directions, i.e., InvariantSA and BO-QSA which improve the slot attention module, DINOSAUR which changes the reconstruction target to DINO features, as well as LSD which introduces the diffusion decoder, to compare with our model. For a fair comparison, apart from the improved component of these methods, the rest components remain consistent with our model. For example, LSD improves the object-centric model with a diffusion decoder, while the backbone network and the slot encoder are the same as our model.

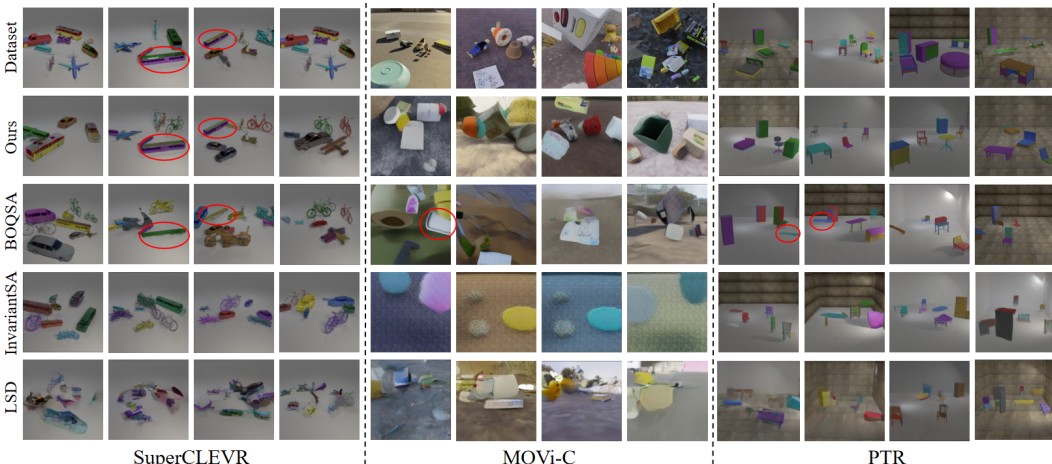

Figure 4: **Compositional generation results**. The first row is images from the datasets, and the rest are the generated samples of models by composing slots extracted from multiple images.

Table 2: **Reconstruction and Compositional generation comparison.** We compare with object-centric models that have generative capability. Lower is better for all the metrics.

| Model | MOVi-C | | | Super-CLEVR | | | PTR | | |
|---|---|---|---|---|---|---|---|---|---|
| | FID | MSE | LPIPS | FID | MSE | LPIPS | FID | MSE | LPIPS |
| BO-QSA (Jia et al., 2023) | 34.96 | 154 | 0.222 | 17.88 | **31** | 0.064 | 16.46 | 40 | 0.098 |
| InvariantSA (Biza et al., 2023) | 163.94 | 484 | 0.421 | 25.15 | 38 | 0.129 | 42.21 | 56 | 0.154 |
| LSD (Jiang et al., 2023) | 69.12 | 661 | 0.404 | 74.68 | 51 | 0.171 | 37.30 | 58 | 0.151 |
| ours | **24.08** | **147** | **0.218** | **15.74** | **31** | **0.059** | **12.98** | **39** | **0.093** |

**Evaluation Benchmarks.** We adopt multiple tasks to evaluate the object-centric representations learned by the model. First, we evaluate the unsupervised object discovery task to test the model's ability to distinguish objects by calculating **ARI-FG, mIoU**, and **mBO** between the ground truth and the object mask predicted by the model. After that, we use the reconstruction metrics **MSE** and **LPIPS** to evaluate the clarity of images generated by the model, as well as introduce the compositional generation task, which randomly combines the slots extracted from the image to generate new images, to evaluate whether the model can separate individual objects and composing them into novel reasonable scenes. The compositional generation task is evaluated by **FID** score between 5000 generated images and the original datasets. Finally, considering that the object-centric model learns through low-level reconstruction signals, we introduce the visual question & answering (VQA) task on Super-CLEVR to examine whether slots can capture high-level semantics about objects. The **answering accuracy** is used as the metric for the VQA task.

## 4.1 OBJECT-CENTRIC REPRESENTATIONS

### 4.1.1 UNSUPERVISED OBJECT DISCOVERY

A basic issue of object-centric learning is the object discovery task that evaluates the one-to-one correspondence between objects and slots. Following previous works, we use ARI-FG, mIoU, and mBO to evaluate how well the masks coincide with objects. Among these metrics, ARI-FG excludes the background pixels during evaluation and tests whether a slot captures a holistic foreground object. mIoU and mBO further demand the model to distinguish between objects and backgrounds.

We show the quantitative results in Tab.1. Our model creates a significant performance gap compared to other models. In MOVi-C, we exceed the previous best score by 6.98, 14.42, and 13.80 according to ARI-FG, mIoU, and mBO. In PTR, the advantage is 4.42, 7.26 and 6.22. In Super-

CLEVR, our ARI-FG outperforms other best models by 16.58, and the boosts of mIoU and mBO are 3.57 and 3.58.

A notable observation is that the reconstruction signal has a great influence on the object representations learned by the models. Our model presents a pattern that reconstructs RGB pixels while constructing constraints on the masks with self-supervised features to optimize object representations, which allows our model to adapt to various datasets and significantly improve performance. By contrast, BO-QSA and InvariantSA reconstruct the raw RGB pixel, leading to high performance on Super-CLEVR, but degrading with the increasing scene complexity. DINOSAUR introduces DINO features as reconstruction targets, which adapt well on MOVi-C, but fail on Super-CLEVR and PTR. Similarly, LSD that introduces VAE code works on MOVi-C and PTR but fails on Super-CLEVR. We further show the visualized comparison in Fig.3, which is consistent with the quantitative results. Our model solves the problem that BOQSA is sensitive to color change on Super-CLEVR and PTR, which tends to divide objects with large inter-color differences into multiple parts. For MOVi-C, there are large object size differences between objects, e.g., the object in the third image in Fig.3 takes up more than half of the image area. Our model is the only one that successfully segments the holistic object. Other models make mistakes on large objects and are often susceptible to secondary factors such as shadows.

### 4.1.2 RECONSTRUCTION AND COMPOSITIONAL GENERATION

A critical property of object-centric models is that the slots are independent and interchangeable, thus allowing for a natural ability to generate novel images by combining slots extracted from given images. Therefore, we further evaluate the object representations with reconstruction and compositional generation tasks. We use two reconstruction metrics, MSE and LPIPS, to evaluate the clarity of generated images, as well as the generation metrics FID to judge whether the model can generate reasonable images through free combinations of slots.

We find that models with better segmentation performance in Tab.1 also generate images with better quality through compositional generation, according to the FID score and reconstruction metrics in Tab.2. The visualization results are shown in Fig.4. Our model presents distinct object properties across all datasets and succeeds in composing new scenes using holistic objects. Other models show inferior results. The appearance of the object in the image generated by LSD is distorted, making it hard to distinguish the objects. InvariantSA succeeds in composing objects on SuperCLEVR, but deteriorates significantly on MOVi-C and only generates meaningless color blocks. BO-QSA generates better results than the first two, but parts of the object appear separately in the generated image in all the datasets, which have been circled in the results of BOQSA in Fig.4. A notable example is the train model in Super-CLEVR, where BOQSA tends to separate the roof and body due to their different colors, resulting in a separate roof appearing in the generated images. Our model succeeds in avoiding this error. For better illustration, we present the original train model on the first line and give the results that our model and BOQSA generated using this model respectively.

### 4.1.3 VISUAL QUESTION & ANSWERING

Previous tasks have proved that our model succeeds in discovering objects and generating novel images with the appearance information in the slot. Here we introduce the visual question & answering (VQA) task to further demonstrate that the quality of object-centric representations is associated with the model's ability to capture high-level semantics. We adopt the ALOE (Ding et al., 2021) structure to accomplish VQA with the slots of each model.

We provide the VQA performance on Super-CLEVR in Tab.3 and divide the questions into three categories, i.e., **count**, **attr** and **judge**, which respectively represent questions with answers are the number of objects that meet the requirements, an attribute of an object, and whether a proposition is correct. Different models show significant performance differences in VQA, and we see a link between VQA and

Table 3: **Super-CLEVR VQA Accuracy.** 'count', 'attr' and 'judge' represents questions with an answer of a number, an attribute, or a judgment.

| Model | Super-CLEVR | | | |
|---|---|---|---|---|
| | count | attr | judge | overall |
| LSD | 49.24 | 36.71 | 61.79 | 49.94 |
| InvariantSA | 43.78 | 54.96 | 63.14 | 55.86 |
| BO-QSA | 51.41 | 57.86 | 65.15 | 59.38 |
| ours | **51.90** | **62.42** | **65.71** | **61.35** |

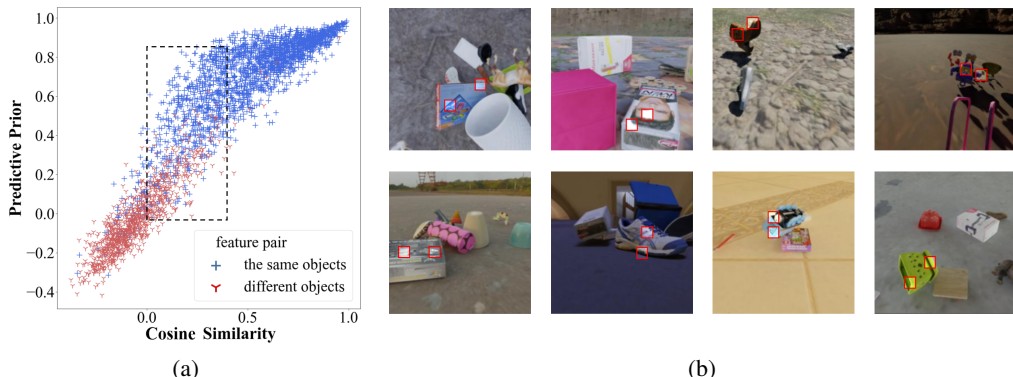

(a)                                     (b)

Figure 5: **Feature pair samples.** **(a)** Cosine similarity and Predictive Prior between feature pairs sampled from MOVi-C. Red and blue represents whether the features come from the same object. In the black dashed box, the Predictive Prior showed better discrimination than cosine similarity. **(b)** Examples of feature pairs with low cosine similarity ($<0.3$) and high Predictive Prior ($>0.5$).

object-centric representation: Models that achieve better performance in object discovery tasks also have higher accuracy in VQA. By breaking down the tasks, we find that the largest performance margin occurs in attribute-related tasks that require the model to find a unique object and answer a property of it, which best embodies the object-centric representations of the models.

## 4.2 ABLATION AND ANALYSIS STUDIES

### 4.2.1 SUPERIORITY OF PREDICTIVE PRIOR

Our work focuses on optimizing the model by exploring the relationships between self-supervised features and constructing constraints on the object masks generated by the model. Compared to Predictive Prior, a more fundamental prior is the cosine similarity between features, which has been widely adopted in segmentation tasks as mentioned in Sec.2. Here we provide a comparison when using similarity-based priors instead of Predictive Prior. Two kinds of priors are considered, i.e. STEGO (Hamilton et al., 2022) and SmoothSeg (Lan et al., 2024). STEGO proposes a distillation method to learn low-rank compact representations from the self-supervised features, while SmooSeg proposes to assign similar labels to patches with similar features. The results are given in Tab.4. In the first row, we provide the performance of the baseline model trained with only reconstruction loss, which is consistent with the performance of BO-QSA in Sec.4.1.1. As is shown in Tab.4, Predictive Prior provides the highest gain across all the datasets.

Table 4: Ablative experiment on the Predictive Prior. We combine object-centric models with priors proposed in previous segmentation research and compare them with Predictive Prior.

| Model | MOVi-C | | | Super-CLEVR | | | PTR | | |
|---|---|---|---|---|---|---|---|---|---|
| | ARI | mIoU | mBO | ARI | mIoU | mBO | ARI | mIoU | mBO |
| baseline | 58.62 | 44.90 | 46.77 | 70.33 | 57.17 | 57.44 | 66.01 | 63.55 | 65.26 |
| baseline + STEGO | 55.24 | 53.38 | 53.97 | 70.18 | 54.22 | 56.22 | 67.36 | 66.62 | 67.45 |
| baseline + SmooSeg | 70.61 | 52.40 | 53.76 | 66.07 | 57.78 | 58.31 | 66.45 | 63.80 | 65.39 |
| baseline + Predictive Prior | **74.80** | **59.32** | **60.47** | **86.91** | **60.74** | **61.02** | **70.43** | **70.81** | **71.48** |

Fig.5 provides further insight into this result. In Fig.5(a) we sample 3000 DINO feature pairs from MOVi-C. Each point represents a feature pair, whose horizontal and vertical coordinates respectively represent the cosine similarity and Predictive Prior between the feature pair. The color of points indicates whether the pair of features comes from the same objects. Red points represent that the features belong to different objects, while blue points represent the same one. Predictive Prior

divides the sample better than cosine similarity. To illustrate the superiority of Predictive Prior, we have marked the region with cosine similarity between 0 and 0.4 with a black dashed box. Samples in this region are difficult to divide by similarity but can be distinguished according to Predictive Prior. In Fig.5(b) we give several samples of feature pairs from the same object with low similarity but high Predictive Prior. Such conditions often occur at the edges, where the features change dramatically, affecting the cosine similarity. In addition, such conditions also occur in objects with complex appearance. For example, on a box printed with a face (the second image in Fig.5(b)), the face part and the box body part share low cosine similarity, but high Predictive Prior.

### 4.2.2 A HEURISTIC FOR SELECTING PROPER THRESHOLD

In our method, the threshold hyperparameter $\tau$ is essential for distinguishing objects. High $\tau$ tends to assign pixels to different slots, and vice versa. Here we propose a simple heuristic to find an appropriate $\tau$ and verify that the model is robust to the value of the threshold.

We first compute the distribution of Predictive Prior between feature pairs from MOVi-C, which is presented as the total length of the columns in Fig.6(a). Note that Predictive Prior is distributed in a bimodal fashion. We further mark the source of feature pairs according to whether the features come from the same or different objects in the ground truth and represent their ratio with the red and blue parts in each column. As a result, they respectively correspond to the higher and lower peaks, which is intuitive because Predictive Prior tends to approach higher values for features from the same object and lower for those from different ones.

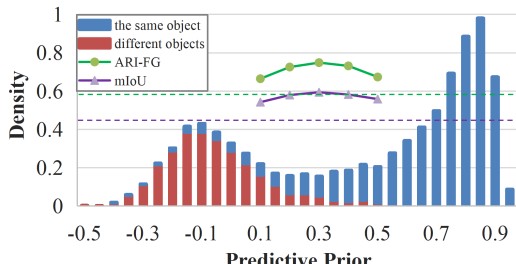

Figure 6: **Heuristic for selecting threshold**. The total length of the columns in the figure represents the distribution density of Predictive Prior, while the red and blue parts represent the ratio of features that belong to different objects or the same object. The lines represent the variation of model performance when the threshold $\tau$ varies in the trough between the two peaks of the distribution. The performances of the baseline model without Predictive Prior are marked with the dashed line of the corresponding color.

Based on this pattern, a heuristic method is to select the point with the lowest Predictive Prior distribution density between the two peaks as the threshold. We verify on multiple datasets that the thresholds determined in this way are all around 0.3, so we choose 0.3 as the threshold for all our models. In addition, we verify the robustness of our model to threshold in Fig.6(b), where we vary the thresholds in the range of [0.1, 0.5] and record model performance using these thresholds. We observe that within [0.2, 0.4], the model's performance only fluctuates by about 2% according to ARI-FG and mIoU. Even when $\tau$ is set to 0.1 or 0.5, the model maintains much higher performance compared to the baseline model without Predictive Prior whose performance is marked with the dashed line. We demonstrate that the model performance is robust to the varying threshold $\tau$.

## 5 CONCLUSION

Current object-centric models struggle with generalizing to complex scenes. We attribute this weakness to the fact that existing models lack effective prior information to identify holistic objects, leading to failure in complex scenes where objects are poorly defined. To address this issue, we draw on human's gestalt ability to construct Predictive Prior based on the intuition that we can infer from one part of an object about other parts. We design a loss function that requires the model to assign the same masks to features that can predict each other and vice versa. Our experiments demonstrate that Predictive Prior significantly improves the model's ability to process objects with the growth of image complexity. On multiple datasets, we show the model's ability to extract individual objects and recompose them into new images while avoiding the problem that object parts may appear in the generated images. In general, our approach uses prediction relationships to construct supervision signals and guide object-centric representations, which we believe is a more successful practice of pushing object-centric learning to complex scene applications.

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
