# Supplementary Material For 'Seeing the part and knowing the whole: Object-Centric Learning with Inter-Feature Prediction'

## A  Further Implementation Details

Here we further elaborate on the network architecture and other configurations of the baseline model.

**Predictive Prior Implementations.** We extract pre-trained features by freezing self-supervised pre-trained Vits (trained-from-scratch MAE for SuperCLEVR and PTR and DINO ViT-S/8 for MOVi-C) and train a prediction network to predict them against each other. Notably, we found that using the key vector from the last attention calculation yielded better results than using the features directly from the last layer of ViT output. We believe that this is because self-supervised pretraining models tend to assign similar attention distributions to the same objects, so their corresponding key vector distributions are more compact and thus easier to predict each other.

**Backbone Network.** We used two backbone networks in different datasets, ResNet34 for Super-CLEVR and PTR, and ViT-S/8 for MOVi-C. Our ResNet-34 model follows previous work (Biza et al., 2023) to replace all the batch-norm layer with group-norm layers. For 128*128 input images, the network first extracts 64-channel, 32*32 resolution features using the stem layer. Then the network sequentially use four sets of residual blocks to extract four intermediate outputs, respectively with shapes 64*32*32, 128*16*16, 256*8*8, and 512*4*4. We use an FPN bottleneck to integrate these outputs and produce a final 64*32*32 output. Finally, position embeddings are added to the output features. Our ViT-S/8 directly use the implementation of Caron et al. (2021). For a 224*224 input image, the ViT encoder outputs features with shape 784*384 (the class token is dropped).

**Slot Encoder.** For SuperCLEVR and PTR, slots have 64 channels. For MOVi-C, slots have 128 channels. We use the official implementation of BOQSA from Jia et al. (2023), as well as the InvariantSA (Biza et al., 2023) implementation from Aydemir.

**Slot Decoder.** For SuperCLEVR and PTR, we use a Spatial Broadcast Decoder Watters et al. (2019) as the slot decoder, where slots are parallelly decoded into RGB reconstructions and alpha masks, which are combined to produce the final reconstruction through alpha blending. Our decoder starts with a learnable position query $\mathbf{P} \in \mathbb{R}^{B \times 256 \times H/16 \times W/16}$, where $B$ is the batch-size and $H, W$ is the height and weight of input images. The decoder processes $\mathbf{P}$ with 4 residual layers, each containing 2 convolution layers with kernel size 3 and a shortcut. Before each residual layer, we bias the query using an adaptive instance normalization layer, where the mean and variance are computed with slots through a two-layer MLP. After each residual layer, we perform a $2\times$ upsample to the feature map and reduce the number of channels by half, thus the decoder outputs a feature map $\mathbf{F} \in \mathbb{R}^{B \times 16 \times H \times W}$ after all the residual layers. Finally, a 1x1 convolution layer reduces the channels to 4, representing the 3-channel RGB reconstructions and 1-channel object masks.

For MOVi-C, we use a transformer-based decoder modified from gansformer Hudson & Zitnick (2022), which also starts from learnable position query $\mathbf{P} \in \mathbb{R}^{B \times 512 \times H/32 \times W/32}$. Slots serve as the latent components and we use 4 cross attention layers to interact between slots and $\mathbf{P}$. After each cross attention layer, $\mathbf{P}$ pass through 2 residual layers and perform an up-sample operation. This result in features with shape $B \times 32 \times H/2 \times W/2$, which finally passes through an up-sample layer and two convolutional layers to provide a $B \times 3 \times H \times W$ reconstruction.

## B  Performance with standard deviation

The object discovery performance of object-centric models is sometimes unstable that the performance of models trained with different seeds varies greatly, so we provide a full performance com-

Table S1: **Full unsupervised object discovery comparison with standard deviation.** Data are represented in the form of 'mean ± std'. The standard deviation is computed with models trained with 3 different seeds.

| Model | MOVi-C | | | Super-CLEVR | | | PTR | | |
|---|---|---|---|---|---|---|---|---|---|
| | ARI | mIoU | mBO | ARI | mIoU | mBO | ARI | mIoU | mBO |
| BO-QSA (Jia et al., 2023) | 58.62 ± 0.84 | 44.90 ± 0.61 | 46.77 ± 0.73 | 70.33 ± 0.73 | 57.17 ± 3.25 | 57.44 ± 3.22 | 66.01 ± 0.21 | 63.55 ± 0.71 | 65.26 ± 0.98 |
| InvariantSA (Biza et al., 2023) | 33.72 ± 3.28 | 26.06 ± 2.45 | 26.94 ± 2.43 | 67.28 ± 0.49 | 58.50 ± 0.48 | 58.86 ± 0.55 | 69.36 ± 0.36 | 33.98 ± 0.28 | 38.28 ± 0.25 |
| DINOSAUR (Seitzer et al., 2022) | 67.82 ± 0.35 | 31.16 ± 1.21 | 38.18 ± 1.39 | 59.52 ± 1.12 | 15.29 ± 2.13 | 15.59 ± 2.24 | 63.80 ± 0.26 | 16.16 ± 1.30 | 17.57 ± 1.42 |
| LSD (Jiang et al., 2023) | 51.98 ± 3.53 | 44.19 ± 0.91 | 45.57 ± 0.80 | 53.05 ± 1.34 | 13.15 ± 1.40 | 13.38 ± 1.23 | 62.22 ± 0.61 | 41.16 ± 4.43 | 41.34 ± 3.49 |
| ours | **74.80 ± 0.80** | **59.32 ± 1.03** | **60.47 ± 1.12** | **86.91 ± 0.23** | **60.74 ± 0.89** | **61.02 ± 1.02** | **70.43 ± 0.74** | **70.81 ± 0.68** | **71.48 ± 0.55** |

parison with Tab.S1, including the standard deviation of the model performance, which is calculated by different models over 3 runs.

## C    FURTHER VISUALIZATION RESULTS

We provide additional generated and object discovery visualizations with Fig.S1, S2 and S3 to demonstrate the effectiveness of our approach. Each image enumerates the model's object discovery results from a dataset. Each row represents the result on an image, the first column is the input image, the second column is the overall segmentation map, followed by the image area occupied by each slot.

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

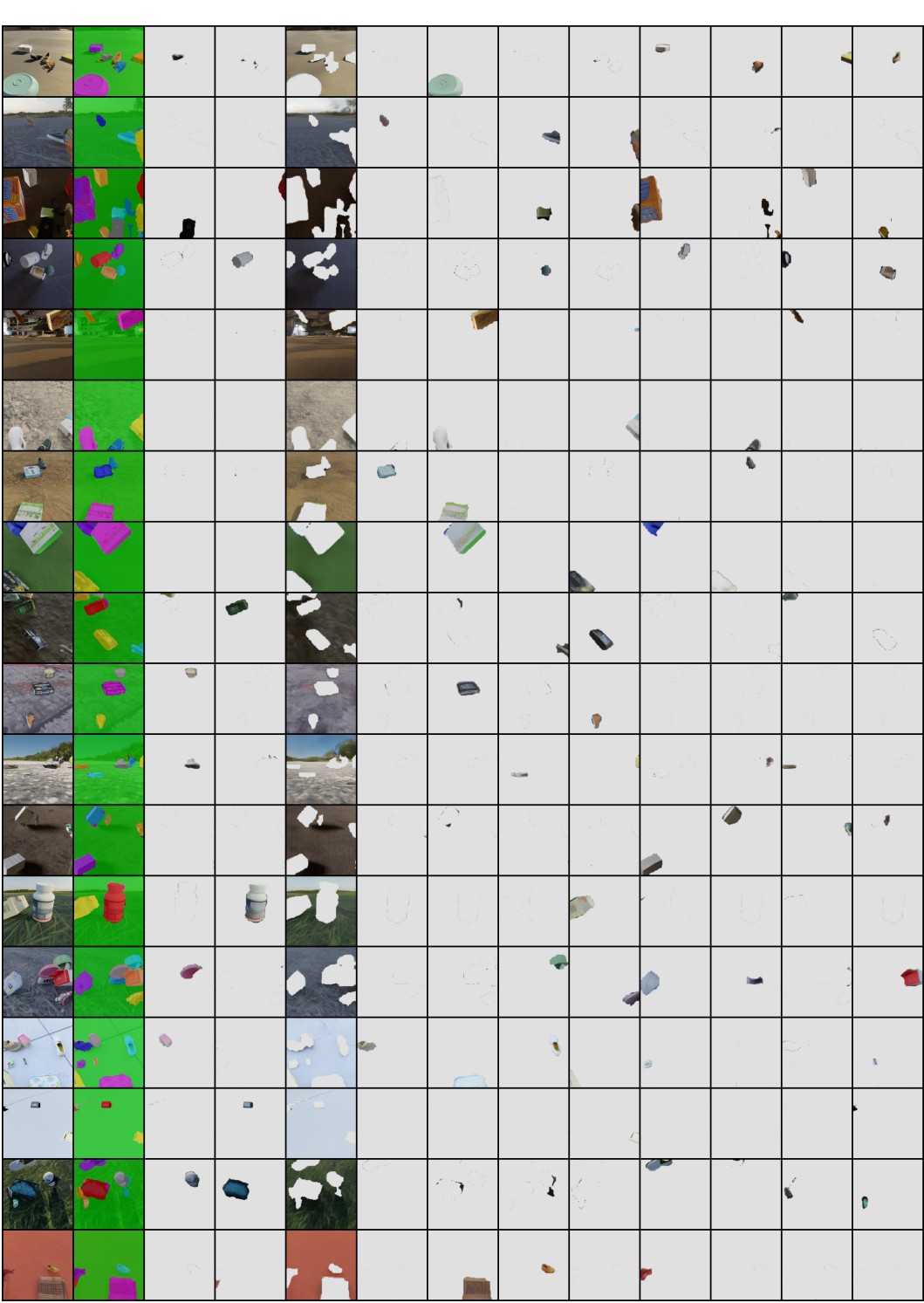

Figure S1: **MOVi visualization results.** The first column is the input image, the second column is the reconstruction result, and the rest are each slot's reconstructed parts.

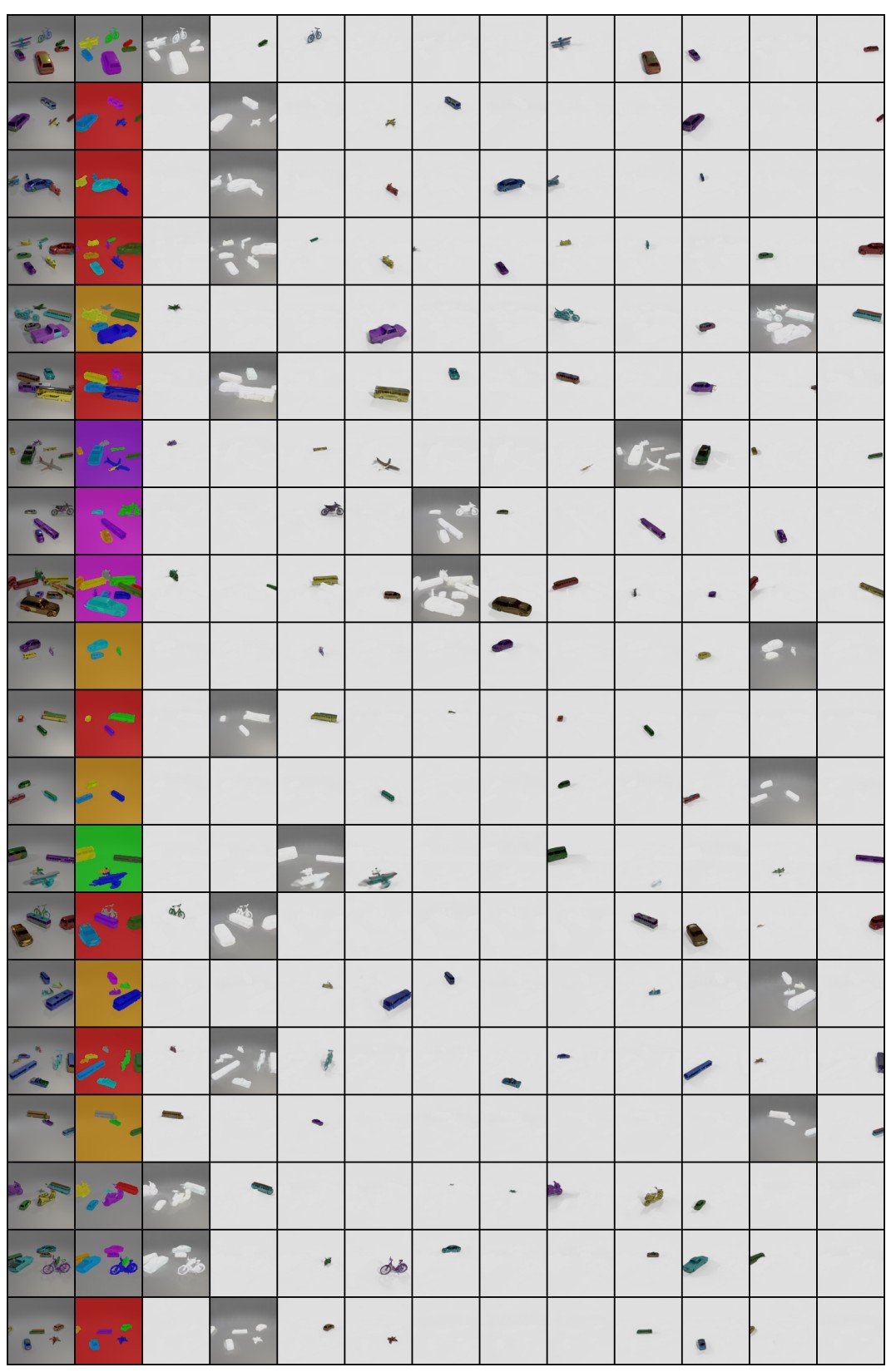

Figure S2: **Super-CLEVR visualization results.** The first column is the input image, the second column is the reconstruction result, and the rest are each slot's reconstructed parts.

Figure S3: **PTR visualization results.** The first column is the input image, the second column is the reconstruction result, and the rest are each slot's reconstructed parts.