# OpenReview forum: "Seeing the part and knowing the whole: Object-Centric Learning with Inter-Feature Prediction"
_ICLR.cc/2025/Conference — ICLR 2025 Conference Withdrawn Submission_

### Official Review · Reviewer_KKuA · 2024-10-29

**Soundness:** 2
**Presentation:** 2
**Contribution:** 2
**Rating:** 3
**Confidence:** 5

**Summary:**

The paper presents a predictor that can forecast the image features at a specific position based on another feature. Additionally, it introduces an object-centric learning method that encourages the assignment of image features, which can effectively predict each other using the pre-trained predictor, to the same slot, and vice versa. Experiments conducted on multiple datasets demonstrate the effectiveness of the proposed method.

**Strengths:**

1. The proposed method is innovative and straightforward to implement.

2. Experiments demonstrate that this method outperforms recent object-centric learning approaches across three datasets.

**Weaknesses:**

1. The proposed predictor does not align with the title "Seeing the part and knowing the whole." In reality, the predictor only observes one part and recognizes another. Additionally, the proposed OCL method does not demonstrate the ability to complete the entire object based on the occluded part. Figure 1 in the paper is also misleading. I would appreciate it if the authors could clarify whether their method can actually complete whole objects from parts or not.

2. The learnable segmentation M is not depicted in Figure 2. I suggest the authors update Figure 2 to include the learnable segmentation M in line 173 around the alpha mask.

3. The presentation of the paper could be improved. The figures are not inserted in PDF format, and some expressions are informal (e.g., the term ‘clamp(a, b)’ in line 251).

4. Compared methods, such as LSD and DINOSAUR, perform better on complex datasets like CLEVRTEX, MOVi-E, and COCO. However, the datasets selected in this paper are relatively simple, raising concerns about the scalability of this approach to more complex data. I suggest authors include their methods on more complex datasets (CLEVRTEX, MOVi-E, and COCO), or disccus their limitation on applying the proposed method to more complex datasets.

5. The rationale for using L1 loss instead of L2 loss as the reconstruction loss is unclear, I would appreciate it if authors could provide their rationale for choosing L1 loss and include an ablation study comparing different loss functions (e.g., L1 vs L2) in their experiments.

**Questions:**

1. According to the attachment, the decoder used for the MOVi-C dataset is transformer-based. How does this type of decoder generate the alpha masks needed to compute the prior loss?

2. Is it possible to apply this method to more complex datasets, as well as images with higher resolutions? Achieving good results on more challenging datasets could enhance the soundness of the proposed method.

3. The visualization in Figure 3 raises some questions, particularly regarding the results of BOQSA on the PTR dataset, which seem to outperform the visualizations in the original paper. Were any techniques implemented to improve the performance?

---

### Official Review · Reviewer_rLea · 2024-11-01

**Soundness:** 2
**Presentation:** 3
**Contribution:** 2
**Rating:** 5
**Confidence:** 5

**Summary:**

This paper proposes a new regularization for constructing the holistic object slot in OCL, which is achieved by utilizing the inter-predictability among the features from different parts of the same object.

**Strengths:**

The writing is good and easy to follow.

**Weaknesses:**

There are some weakness and questions that required to be answered clearly:

1) For the gestalt ability of our human, are the different parts naturally predicted according to the appearance/structure? Or, human predict the missing parts given the prior that we already have a semantic understanding? Any proof?

2) For the inter-predictability among the features, how to deal with the semantic/object occurance in the real world? For example, given the high occurance of the keyboard and mouse, their feature could be with high inter-predictability. How to limit the inter-predictability on the component/part level?

3) For the prediction of similarity, relative postion should be used. The utilization of absolute position is wrong, which cannot reveal the structural information among the parts. Moreover, it will leads to lots of noise for semantic understanding, since object semantics are postion invariant.

4) The training of the similarity prediction seems require supervision, which is unfair for other unsupervised methods.

5) Given the training loss for similarity prediction only focuses on increasing the cosine similarity, any possible that a trival solution will appear, which predict high similarity for any pair of features.

**Questions:**

Please refer to the weaknesses.

---

### Official Review · Reviewer_V4gp · 2024-11-02

**Soundness:** 2
**Presentation:** 3
**Contribution:** 2
**Rating:** 5
**Confidence:** 4

**Summary:**

Humans instinctively decompose scenes into objects, enabling strong visual understanding. Object-Centric Learning (OCL) seeks to encode scene information into object vectors called ‘slots.’ Traditional OCL models use an auto-encoding approach, reconstructing images from these slots but often fail with complex objects, as reconstruction alone doesn’t ensure accurate object grouping. To improve this, this paper introduces a Predictive Prior inspired by human gestalt perception, where features of the same object can predict each other. This prior is implemented as an external loss, guiding the model to group predictable features into the same slot and separate those that aren't. The paper shows decent results on SuperCLEVER, MoVI-C etc.

**Strengths:**

1) Overall intuition of the paper makes sense and representing part and whole of a scene is a pretty important problem as discussed in [1] and in the literature multiple times.
2) Results on SuperCLEVER, MoVI-C are decent and show the efficacy of the current method pretty well.

**Weaknesses:**

1) Results are missing on real world datasets like COCO & OpenImages. The current results are on CLEVEr and MoVI-C which are not very representative of real world results.
2) Comparison with diffusion based approaches like SlotDiffuzr[1], SysBinder[2] is missing. Adding diffusion based method will be pretty critical to the paper.
3) Computational cost in adding the PREDICTIVE PRIOR? It would be good to see the computational cost added by newer modules introduced in the paper.
4) Discussion on [3] should definitely be added in the paper.
References:
1) SlotDiffuzr: SlotDiffusion: Object-Centric Generative Modeling with Diffusion Models
2) NEURAL SYSTEMATIC BINDER
3) How to represent part-whole hierarchies in a neural network.

**Questions:**

Overall the paper is good, but the results are missing on large scale real-world datasets which is definitely a issue in the current version.

---

### Official Review · Reviewer_y5Yh · 2024-11-03

**Soundness:** 3
**Presentation:** 3
**Contribution:** 3
**Rating:** 6
**Confidence:** 4

**Summary:**

The paper introduces an interesting idea to object centric learning (ocl) called Predictive Prior, inspired by human perception abilities. Traditional ocl models use an auto-encoding paradigm to create object representations by assigning image features to discrete object "slots" and reconstructing images from these slots. However, these models struggle with complex object appearances due to reliance on color or spatial regularities.

The Predictive Prior approach leverages the principle that features belonging to the same object can predict each other. It trains a prediction network to assess the mutual predictability between features across different spatial locations within an image. This prediction-based relationship is then used to guide object-slot assignments.

Experiments on datasets such as MOVi-C, Super-CLEVR, and PTR show that the Predictive Prior-based model outperforms previous OCL methods in object discovery, compositional generation, and visual question answering (VQA).

**Strengths:**

- The work shows strong results across various datasets and baselines.
- The idea is interesting and is well analyzed.
- Clean writing and figures

**Weaknesses:**

- Missing Ablations, There are multiple loss functions used, however they haven't been ablated
- Certain SoTA baselines on Dino for unsupervised segmentation are missing such as : CuTLER (https://arxiv.org/pdf/2301.11320https://arxiv.org/pdf/2301.11320)

**Questions:**

- Can the authors compare or discuss results against methods such as CuTLER that use dino features and get good results.
- How much role does pre-trained Dino play in the improvement across baselines. What if the authors trained from scratch using the new objective.
- Can the authors ablate reconstruction vs their proposed objective, how does switching of one of them affect final accuracy?

---

### Note · Authors · 2024-11-15

I have read and agree with the venue's withdrawal policy on behalf of myself and my co-authors.